# Food insecurity among African Americans in the United States: A scoping review

**Elizabeth Dennard**[1]*, **Elizabeth Kristjansson**[2], **Nedelina Tchangalova**[3], **Sarah Totton**[4], **Donna Winham**[5], **Annette O'Connor**[6]

**1** Office of Applied Research and Safety Assessment, Food and Drug Administration, Laurel, Maryland, United States of America, **2** School of Psychology, University of Ottawa, Ottawa, Ontario, Canada, **3** STEM Library, University of Maryland, College Park, Maryland, United States of America, **4** Department of Population Medicine, University of Guelph, Guelph, Ontario, Canada, **5** Food Science and Human Nutrition, Iowa State University, Ames, Iowa, United States of America, **6** College of Veterinary Medicine, Michigan State University, East Lansing, Michigan, United States of America

☯ These authors contributed equally to this work.
* Elizabeth.dennard@fda.hhs.gov

**Data Availability Statement:** All relevant data are within the paper and its Supporting information.

**Funding:** The authors received no specific funding for this work.

## Abstract

In 2019, the estimated prevalence of food insecurity for Black non-Hispanic households was higher than the national average due to health disparities exacerbated by forms of racial discrimination. During the COVID-19 pandemic, Black households have experienced higher rates of food insecurity when compared to other populations in the United States. The primary objectives of this review were to identify which risk factors have been investigated for an association with food insecurity, describe how food insecurity is measured across studies that have evaluated this outcome among African Americans, and determine which dimensions of food security (food accessibility, availability, and utilization) are captured by risk factors studied by authors. Food insecurity related studies were identified through a search of Google Scholar, PubMed, CINAHL Plus, MEDLINE®, PsycINFO, Health Source: Nursing/Academic Edition, and Web of Science™ (Clarivate), on May 20, 2021. Eligible studies were primary research studies, with a concurrent comparison group, published in English between 1995 and 2021. Ninety-eight relevant studies were included for data charting with 37 unique measurement tools, 115 risk factors, and 93 possible consequences of food insecurity identified. Few studies examined factors linked to racial discrimination, behaviour, or risk factors that mapped to the food availability dimension of food security. Infrequently studied factors, such as lifetime racial discrimination, socioeconomic status (SES), and income insecurity need further investigation while frequently studied factors such as age, education, race/ethnicity, and gender need to be summarized using a systematic review approach so that risk factor impact can be better assessed. Risk factors linked to racial discrimination and food insecurity need to be better understood in order to minimize health disparities among African American adults during the COVID-19 pandemic and beyond.

**Competing interests:** The authors have declared that no competing interests exist.

## Introduction

### Description of the problem

As of 2019, 10.5% of United States (US) households (13.7 million households) experienced food insecurity and 4.1% of these households (5.3 million households) experienced very low food security at some time during the year [1]. Rates of food insecurity were significantly higher than the national average for households with Black, non-Hispanic, household reference persons (19.1 percent) [1]. Households that experience food insecurity lack access to enough food for an active and healthy lifestyle for all household members [2]. The COVID-19 pandemic has caused a public health and economic crisis with repercussions that have led to an increase in the number of people experiencing food insecurity. In 2020, African Americans experienced more negative health outcomes linked to COVID-19, the disease caused by SARS-CoV-2, than other populations due to a combination of factors including racial discrimination, disparities linked to income and health, and inconsistent access to food [2]. Further, in the United States, individual studies have reported that African American households are two to three times as likely to experience consistent food insecurity when compared to the general population [3–5] These prior findings indicate that race is associated with food insecurity. However, many individual- and group-level factors other than race have been investigated for an association with food insecurity. A comprehensive list of studied risk factors and their relationship to food insecurity among African American households is not available. A comprehensive list is needed to understand which relationships exist and which intervention opportunities need to be investigated. Diverse metrics of food security have been employed by numerous authors across the literature. According to Ashby and colleagues [6], "accurate measurement of food insecurity is imperative to understand the magnitude of the issue and to identify specific areas of need, in order to effectively tailor policies and interventions for its alleviation." To understand the implications of current study findings, each citation and corresponding findings must be placed in the context of other studies that assess food insecurity among African American adults in the United States.

### Objectives

The first objective of this review was to identify factors that have been investigated for an association with food insecurity among African American adults across the peer-reviewed literature. Knowledge of these factors will identify critical research gaps and highlight areas for future research. The second objective was to describe how food insecurity has been measured in studies that have evaluated this outcome among African American populations. Knowledge of food security metrics will identify how comparable current measures and potential findings are across the literature. The final objective was to map each risk factor identified or considered by researchers to the three primary dimensions of food security (food accessibility, availability, and utilization) to identify potential gaps across each dimension. Table 1 serves as a glossary of terms and definitions for food security and relevant proxy variables.

## Materials and methods

### Protocol and registration

Registering a protocol for systematic reviews in advance promotes transparency, reduces bias, and eliminates unintended duplication of effort [7, 8]. The PRISMA checklist was developed by a 24-member expert panel following published guidance and contains 22 reporting items to help readers develop a greater understanding of relevant terminology, concepts, and key items to report for scoping reviews [9]. The protocol followed the framework set by Munn et al.

**Table 1. Glossary of food security terms.**

| Term | Definition |
|---|---|
| food security | Food security refers to access by all people at all times to enough food for an active and healthy lifestyle [1]. |
| food insecurity | Households that experience food insecurity lack access to enough food for an active and healthy lifestyle for all household members [2]. |
| food availability | Food availability refers to a reliable and consistent source of enough quality food for an active and healthy lifestyle (environmental factors) [6]. |
| food accessibility | Food accessibility acknowledges the resources required in order to obtain and put food on the table (economic factors) [6]. |
| food utilization | Food utilization refers to the intake of safe food and the human resources required to transform food into meals [6]. |
| food stability | Food stability can be achieved when all three domains (availability, accessibility, and utilization) become sustainable over time [6]. |

(2018) and Arksey and O'Malley (2005) [10, 11], as well as the guidelines in protocol was drafted using the PRISMA Extension for Scoping Reviews (PRISMA-ScR): Checklist and Explanation. The protocol was registered with the Systematic Reviews for Animals and Food (SYREAF) on December 30, 2019 (https://syreaf.org/wp-content/uploads/2022/05/Scoping-Review-Protocol_Signed.pdf). The methodology was informed by Munn et al. (2018)'s guidance and Arksey and O'Malley (2005)'s framework [10, 11].

## Eligibility criteria

The eligibility criteria for study inclusion were defined based on the population (P)—adult African Americans, and the outcome (O)—food insecurity. Peer-reviewed articles published in English between 1995–2021 were eligible for inclusion in this paper.

## Eligible study designs

Eligible studies were primary research studies with a concurrent comparison group: observational studies (cross-sectional, cohort, and case control), randomized controlled trials, and primary research studies that evaluated risk factors between time periods (before and after). Studies that assessed interventions were also included.

## Eligible participants

Relevant participants were African American adults, 18 to 64 years of age, living in the United States. If a study contained a subset of a sample that matched the population of interest, the subset of participants was included if data was reported separately. One possible source of ambiguity among identified citations included the definition and use of the term "African American" in the literature. The United States Census Bureau adheres to the 1997 Office of Management Budget (OMB) standards on race and ethnicity, which includes five categories: Asian, Black or African American, Native Hawaiian or Pacific Islander, American Indian or Alaska Native, and White [12]. According to Rastogi and colleagues, "The Black racial category includes people who marked the 'Black or African American' checkbox. It also includes respondents who reported entries such as African American; Sub-Saharan African entries, such as Kenyan and Nigerian; and Afro-Caribbean entries, such as Haitian and Jamaican" [13]. The category for Black and African American people serves as a broad descriptor for study participants who may not share the same ethnicity, culture, or immigration status. Rastogi and colleagues explain further that "these federal standards mandate that race and

Hispanic origin (ethnicity) are separate and distinct concepts and that when collecting these data via self-identification, two different questions must be used" [13]. This distinction between race and ethnicity is relevant to this scoping review because the intention was to include study participants who only identify themselves as African American. Immigration status is another key factor that may have impacted the eligible study population of interest. For this scoping review, citations were excluded if the researcher's study population of interest comprised only immigrants or refugees.

## Eligible outcomes

The outcome of interest was food insecurity. Some authors may have used the following terms to describe food insecurity: food availability, food accessibility, food utilization, food supply, food intake, undernourishment, food deprivation, hunger, malnutrition, and use of food assistance programs. These proxy variables of food insecurity were also eligible for inclusion in this study.

## Search sources

The search for relevant studies was conducted in six databases: PubMed (US National Library of Medicine), EBSCO databases (CINAHL Plus, MEDLINE®, PsycINFO, Health Source: Nursing/Academic Edition), and Web of Science™ (Clarivate) on May 20, 2021. Both MEDLINE (EBSCO) and legacy PubMed, the old interface, were searched due to the variations of the database syntax and features. In addition to the databases above, Google Scholar was searched to find additional studies that may have been missed through the database searches. Relevant full-text publications were obtained through available subscriptions through the University of Maryland, University of Guelph, and Iowa State University Libraries. Reference lists of the included primary research articles and retrieved systematic reviews were examined to identify any relevant publications. DistillerSR® (Evidence Partners, Ottawa, Canada) software was used for article screening and data extraction.

## Search strategy

The search strategy was designed by a public health librarian in consultation with other team members. The search strategy was checked for comprehensiveness and errors against the *PRESS Peer Review of Electronic Search Strategies Guidelines* [14]. Search strategies for each database and corresponding results are shown in S1 Appendix (S1–S3 Tables). Results were restricted to publication year 1995–2021, English language, and peer-reviewed publications. The US Department of Agriculture (USDA) began collecting data annually regarding food access, food spending, and sources of food assistance in the United States in 1995 [15]. Therefore, this regulatory activity represents a reasonable starting point for relevant studies to be included in this paper.

## Selection of sources

Search results were uploaded into EndNote X9 Desktop and duplicate records removed. Title/abstract screening, full-text screening, and data extraction were independently performed by two authors in DistillerSR®. Both reviewers received training prior to the screening process using piloted forms and discussion until agreement about interpretation was reached. The title/abstract screening form was piloted with 100 records while the full-text screening form was piloted with five records. Conflicts were resolved through discussion until consensus was

reached based on detailed justifications provided by each reviewer. The screening forms are included in S3 Appendix.

## Data charting and analysis

Data charting forms were developed and reviewed to determine study characteristics and data items for extraction. Two reviewers independently captured data items, discussed findings, and updated all forms as changes were made. Data extraction forms are included in S3 Appendix.

### Data items and extraction

Data extraction captured general study characteristics, study population characteristics (state, region, age distribution, and number of participants), study design, exposures investigated, and relevant measures. These food insecurity metrics might be used at the individual level to represent the experiences, behaviours, or conditions of an individual or a single household [1]. Alternatively, these metrics might be aggregated to represent a group at the ecological or group level. For example, a study might report the proportion of households in a region that skip meals more than twice in one week or the proportion of households in a neighbourhood with a cut-off listed in the USDA (2018)'s 18-item questionnaire. For this scoping review, all measures of food security described in the literature were extracted.

### Risk of bias and study quality

The authors did not assess risk of bias or study quality of the included studies, as risk-of-bias assessment is not required for scoping reviews [10]. According to Munn and colleagues (2018) "as scoping reviews do not aim to produce a critically appraised and synthesized result/answer to a particular question, an assessment of methodological limitations or risk of bias of the evidence included within a scoping review is generally not performed unless there is a specific requirement due to the nature of a scoping review aim" [10].

### Critical appraisal of individual sources of evidence

A critical appraisal of the included studies was not conducted, consistent with Arksey and O'Malley (2005)'s guidance [11].

### Synthesis of results

After data extraction, the factors were mapped to no more than three of the four unique dimensions of food security: food availability, food accessibility, and food utilization. Table 1 provides definitions of these proxy variables of food insecurity. The extracted risk factors were also mapped as being at the individual or group level and whether a risk factor appeared to be a "cause" or "possible consequence" of food insecurity. If a risk factor identified in the study served as a "possible consequence" of food insecurity, this term was not categorized into the food security dimensions (food availability, accessibility, and utilization) for risk factors. For example, a study participant's mental health status or "depression score" could serve as both a "cause" of food insecurity due to lack of food accessibility or it could serve as a "consequence" of experiencing food insecurity due to lack of food utilization. If the risk factor fell into the "cause" category only, the factor was categorized based on the three food security dimensions described above. Finally, these variables were placed into ten descriptive categories: demographic (individual characteristics such as age and sex), household (marital status and single parent status), economic (household income and family poverty), behavioural (lifestyle habits,

actions, and behaviours), nutritional, physical environment (physical, chemical, and biological factors external to a person), social environment (social factors external to a person), physical health (physical and genetic health factors), mental health, and COVID-19 related risk factors. This process was completed by two reviewers and then conflicts were resolved through discussion to ensure consistent classification.

# Results

## Selection of citations

The results of the search and eligibility screening process are presented in Fig 1 [16].

## Characteristics of included studies

The characteristics of included studies are described in S2 Appendix (S4–S6 Tables). which provides an overview of food security measures described by authors, citation characteristics (state, region, and study design), and study population characteristics (spread of ages, participant count, and household count).

## Synthesis of results

Data were extracted from ninety-eight citations. Seventy-three studies employed a cross-sectional design, while the remaining studies implemented the following study designs: cohort/longitudinal (n = 19 studies), randomized controlled trial (n = 3 studies), qualitative (n = 2 studies), and concept mapping (n = 1 study). Studies were conducted in multiple states, but many authors did not report a specific state (n = 35). Twenty-eight studies reported findings from urban, both urban and rural (n = 12 studies), and rural (n = 3 studies) regions while the remaining studies did not report a specific region (n = 55 studies).

For the 115 risk factors identified, demographic characteristics represented the majority of factors described in the literature (n = 53 factors). Behavioural (lifestyle and nutritional habits, n = 5 factors), environmental (physical and social environment, n = 38 factors), health-related characteristics (physical and mental health, n = 15 factors), and COVID-19 related risk factors (n = 4) were less commonly reported. For possible consequences of food insecurity (n = 92 factors), the following terms received the greatest number of hits across the reviewed citations: self-reported health status (n = 16 citation hits), total number of people in household (n = 14 citation hits), SNAP recipient (n = 14 citation hits), depression or depressive symptoms (n = 12 citation hits), and body mass index (BMI) (n = 8 citation hits). The results of the risk factor mapping process are presented in Fig 2.

The 115 risk factors were mapped to five broad categories (demographic, behaviour, environment, health-related factors, and COVID-19 related factors) along with ten descriptive subcategories for further risk factor categorization. Each subcategory was further mapped to the three dimensions of food security (food accessibility, availability, and utilization) and each combination available (1. Accessibility and Availability; 2. All Categories; 3. Accessibility; 4. Accessibility and Utilization; 5. Availability (Fig 2). None of the identified risk factors mapped to food utilization exclusively, so this category was not represented in the figure. Demographic factors mapped most frequently to the accessibility category while household and economic factors mapped to the food accessibility and utilization categories. Behavioural factors linked to behaviour and nutrition mapped to all three dimensions of food security, while COVID-19 related factors and health-related factors primarily mapped to food accessibility and utilization. Most of the physical environmental factors mapped to food accessibility and availability, while most social environmental factors mapped to food accessibility exclusively. Ultimately,

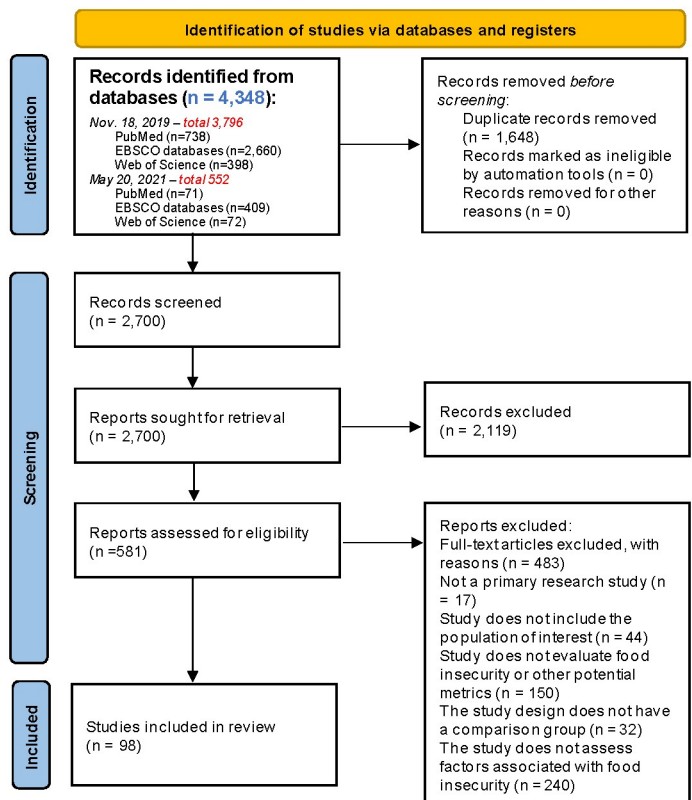

**Fig 1. PRISMA 2020 flow diagram for new systematic reviews which included searches of databases and registers only.** This diagram depicts the flow of information through different phases of a scoping review and maps the number of records identified, included and excluded, and exclusion justifications.

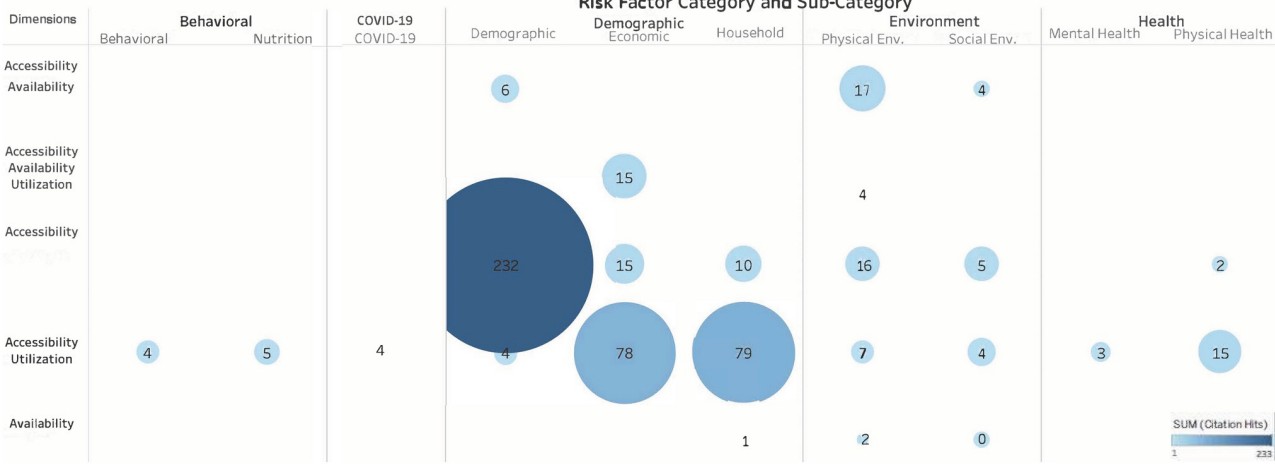

**Fig 2. Dimensions of food insecurity Evidence and Gap Map (EGM).** This diagram depicts the sum of citation hits (1–233) per risk factor category (behavioural, COVID-19, demographic, environment, and health) and how each category is mapped to the dimensions of food insecurity (accessibility, availability, and utilization).

this scoping review provides a visual breakdown of risk factor categorization across each dimension and possible combination of food security in all included studies.

Thirty-seven measures of food security were identified across 98 citations. Most authors implemented the U.S. Household Food Security Survey Module (n = 16), the Six-Item Short Form of the Food Security Survey Module (n = 16), and the Eighteen-Item Household Food Security Scale (n = 13). The remaining studies referenced other measures of food security. Adaptations of the USDA Food Security Survey Module included the US Adult Food Security Survey Module, a 2-item screener derived from the 18-Item US Household Food Security Screen, and a 3-item adaptation from the USDA Food Insecurity Scale [17–21]. Non-USDA metrics included the National Health Interview Survey on Disability, the 2007 AIDS Alabama Needs Assessment Survey, the Survey of Income and Program Participation (SIPP), the Food Insufficiency Indicator (from SEED OK Survey), the Current Population Survey Food Security Supplement (CPS-FSS), the Health and Retirement Study (HRS) Food Insecurity Questionnaire, the Radimer-Cornell Hunger and Food Insecurity Instrument, the Access to Healthy Foods Scale, and the NHANES Food Security Module [22–31]. Remaining metrics include Food Sufficiency Status based on four self-reported risk situations that were related to absence of food and forced scarce-resource decisions, neighbourhood supermarket density per 10,000 people, receipt of food stamps in the past 12 months, the number of full-service retail food outlets (RFOs) in the neighbourhood, and WIC receipt [32–35].

Most of the demographic factors (n = 53 risk factors), including household and economic terms, were mapped to the food access category (n = 52 risk factors) while remaining dimensions of food security, food availability (n = 5 risk factors) and food utilization (n = 26 risk factors), were mapped less frequently (Table 2). Examples of identified demographic risk factors include age, race/ethnicity, gender, number of children in household, socioeconomic status (SES), and family poverty. All behavioural factors (n = 5), including lifestyle habits and terms linked to nutrition, mapped to food access and food utilization (Table 3). Most of the environmental factors (n = 38 factors), including physical and social environment terms, mapped to the food access category (n = 36 factors), while food availability (n = 19 factors) and food utilization (n = 10 factors), were mapped less frequently (Table 4). Examples of identified environmental risk factors include geographic location, living situation, neighbourhood grocery store availability, and neighbourhood safety from crime and violence. All health-related factors (n = 15), including physical and mental health terms, mapped to the food access dimension of food insecurity. Most of these terms also mapped to the food utilization category (n = 13) while none of them mapped to food availability (Table 5). Examples of identified health-related risk factors include human immunodeficiency virus (HIV) status, arthritis, alcoholism, liver fibrosis, and health insurance status. All COVID-19 related risk factors (n = 4), including impact of COVID-19 on employment, stay-at-home orders, decreased income due to COVID-19, and unemployed prior to pandemic, mapped to the food access and utilization dimension of food security (Table 6).

## Discussion

### Summary of the evidence

The findings from this scoping review suggest that a wide range of risk factors have been evaluated for an association with food insecurity among African American adults across the peer-reviewed literature. The demographic (n = 53 risk factors) and environmental (n = 38 risk factors) categories represented the greatest number of risk factors evaluated across studies, which suggests that these categories, and relevant terms within each group, have received more

**Table 2. Demographic risk factors mapped to the dimensions of food security.**

| Term | Citation Hits | Sub Category | Accessibility | Availability | Utilization | Level |
|---|---|---|---|---|---|---|
| Race/ethnicity | 65 | Demographic | Accessibility | | | Individual |
| Age | 54 | Demographic | Accessibility | | | Individual |
| Education | 52 | Demographic | Accessibility | | | Individual |
| Gender (social) | 41 | Demographic | Accessibility | | | Individual |
| Household income | 29 | Economic | Accessibility | | Utilization | Group |
| Employed/Unemployed | 28 | Economic | Accessibility | | Utilization | Individual |
| Marital status (partnered status) | 28 | Household | Accessibility | | Utilization | Group |
| Number of children in household | 20 | Household | Accessibility | | Utilization | Group |
| Income | 15 | Economic | Accessibility | | Utilization | Individual |
| Family poverty | 11 | Economic | Accessibility | Availability | Utilization | Group |
| Child's age | 10 | Household | Accessibility | | Utilization | Group |
| Race | 10 | Demographic | Accessibility | | | Individual |
| Single parent (status) | 6 | Household | Accessibility | | Utilization | Group |
| Time (year) | 6 | Demographic | Accessibility | Availability | | Group |
| Mother's age | 5 | Household | Accessibility | | Utilization | Group |
| Child's gender | 4 | Household | Accessibility | | | Individual |
| Female-headed household | 4 | Household | Accessibility | | | Group |
| Home ownership | 4 | Economic | Accessibility | | Utilization | Individual |
| Documentation status (work permit, citizen, legal permanent resident, etc.) | 3 | Demographic | Accessibility | | | Individual |
| Poverty rate | 3 | Economic | Accessibility | Availability | Utilization | Group |
| Sexual orientation | 3 | Demographic | Accessibility | | | Individual |
| Unemployment rate | 3 | Economic | Accessibility | | | Group |
| Disability | 2 | Demographic | Accessibility | | Utilization | Individual |
| Family monthly poverty level index | 2 | Economic | Accessibility | | | Group |
| History of military service | 2 | Demographic | Accessibility | | | Individual |
| Hours of work | 2 | Economic | Accessibility | | Utilization | Individual |
| Infant/child race/ethnicity | 2 | Household | Accessibility | | | Individual |
| Maternal union transitions | 2 | Economic | Accessibility | | | Individual |
| Pregnant woman (pregnancy status) | 2 | Demographic | Accessibility | | Utilization | Individual |
| Baby's father in household | 1 | Household | Accessibility | | Utilization | Group |
| Baby's grandmother in household | 1 | Household | Accessibility | | Utilization | Group |
| Bank account ownership | 1 | Economic | Accessibility | | | Individual |
| Child in household on NSLP (National School Lunch Program) | 1 | Household | | Availability | | Both |
| Credit card ownership | 1 | Economic | Accessibility | | | Individual |
| Disabled adults in household | 1 | Household | Accessibility | | Utilization | Group |
| Disabled child in household (not receiving SSI) | 1 | Household | Accessibility | | Utilization | Group |
| Disabled child in household (receiving SSI) | 1 | Household | Accessibility | | Utilization | Group |
| English proficiency | 1 | Demographic | Accessibility | | | Individual |
| Financial capability | 1 | Economic | Accessibility | | Utilization | Both |
| Financial hardship from medical bills | 1 | Economic | Accessibility | | | Both |
| Gender modality (transgender or cisgender) | 1 | Demographic | Accessibility | | | Individual |
| Has dependents | 1 | Household | Accessibility | | Utilization | Individual |
| Have enough money to buy food at the hospital | 1 | Economic | Accessibility | | | Individual |
| History of incarceration | 1 | Household | Accessibility | | Utilization | Both |
| Income insecurity | 1 | Economic | Accessibility | Availability | Utilization | Both |
| Parental drug use | 1 | Household | Accessibility | | Utilization | Group |
| Parental incarceration | 1 | Household | Accessibility | | Utilization | Group |

(*Continued*)

8</PLOS ONE

Food insecurity among African Americans in the United States

**Table 2.** (Continued)

| Term | Citation Hits | Sub Category | Accessibility | Availability | Utilization | Level |
|---|---|---|---|---|---|---|
| Religion | 1 | Demographic | Accessibility | | | Individual |
| Senior in household | 1 | Household | Accessibility | | Utilization | Group |
| Socio-economic status (SES) | 1 | Economic | Accessibility | | | Individual |
| State welfare expenditures | 1 | Economic | Accessibility | | | Group |
| Unexpected expenses | 1 | Economic | Accessibility | | | Individual |
| Will lose income from your job because of hospital stay | 1 | Economic | Accessibility | | | Individual |

representation when compared to other categories (behavioural, health-related, and COVID-19-related categories).

COVID-19 related factors (n = 4), behavioural factors (n = 5), and health related factors (n = 15) comprised the fewest number of risk factors across included studies. This serves as a significant data gap compared to demographic and environmental characteristics, because these sub-categories have received less attention by authors. In future studies, it is critical for researchers to consider risk factor representation across the literature by examining behavioural and health-related risk factors among African American adults to fill current data gaps. A few examples include sexual orientation [22], English proficiency [34], pregnancy status [36, 37], religion [38], lifetime racial discrimination [18], neighbourhood safety from crime and violence [26], neighbourhood grocery store availability [38], impairment that limits use of public transportation [24, 39], HIV status [40], decreased income due to COVID-19 [41], the impact of COVID-19 on employment, and stay-at-home orders [42]. Future primary research studies could focus on these under-represented risk factors that may perpetuate food insecurity among African American adults instead of examining risk factors that have been extensively evaluated by other researchers. Authors should also consider findings from multiple publications, including similar studies, scoping reviews, and systematic reviews, instead of formulating hypotheses based on a single finding or publication. The inference obtained from a single publication is limited; therefore, authors of future studies should consider findings from multiple studies to refine metrics and improve study design for stronger inference about described associations.

Diverse metrics of food security (n = 37 metrics) have been employed across this body of included studies to measure a single outcome. The use of multiple measures for a single outcome presents issues for understanding the entire body of work available to readers. If researchers and clinicians are willing to modify standardized measures of food security, then a justification for this modification must be reported. For example, the 2-item screen derived from the 18-Item US Household Food Security Screen could impact the accuracy of the measurement of food insecurity. In addition, it is important for researchers and clinicians to consider the value of individual questions within modified screeners. Variation in questions and

**Table 3.** Behavioural risk factors mapped to the dimensions of food security.

| Term | Citation Hits | Sub Category | Accessibility | Availability | Utilization | Level |
|---|---|---|---|---|---|---|
| Drug problem | 3 | Behavioral | Accessibility | | Utilization | Individual |
| "I'm too busy to take the time to prepare healthy foods" | 2 | Nutrition | Accessibility | | Utilization | Individual |
| SNAP receipt in past year | 2 | Nutrition | Accessibility | | Utilization | Individual |
| Time since SNAP distribution | 1 | Nutrition | Accessibility | | Utilization | Individual |
| Taking prescribed medications | 1 | Behavioral | Accessibility | | Utilization | Individual |

**Table 4. Environmental risk factors mapped to the dimensions of food security.**

| Term | Citation Hits | Sub Category | Accessibility | Availability | Utilization | Level |
|---|---|---|---|---|---|---|
| Urbanicity | 7 | Physical Environment | Accessibility | Availability | | Group |
| Access to car | 5 | Physical Environment | Accessibility | | | Both |
| Living situation (living alone vs with spouse/family/room-mates) | 4 | Physical Environment | Accessibility | | Utilization | Both |
| Social support (to borrow money from) | 4 | Social Environment | Accessibility | | | Individual |
| Access to help from family, friends, neighbors | 3 | Social Environment | Accessibility | | Utilization | Individual |
| Geographic location | 2 | Physical Environment | Accessibility | Availability | | Group |
| State | 2 | Physical Environment | Accessibility | Availability | | Group |
| Social capital | 2 | Social Environment | Accessibility | | | Individual |
| Metropolitan residency | 1 | Physical Environment | Accessibility | Availability | | Group |
| Fruit and vegetable selection in neighborhood | 1 | Physical Environment | Accessibility | Availability | | Group |
| Have transportation to get food while at the hospital | 1 | Physical Environment | Accessibility | | | Individual |
| Neighborhood aesthetic quality | 1 | Physical Environment | Accessibility | Availability | | Group |
| Neighborhood walking/exercise environment | 1 | Physical Environment | Accessibility | | | Group |
| Neighborhood safety from crime/violence | 1 | Physical Environment | Accessibility | Availability | | Group |
| Neighborhood grocery store availability | 1 | Physical Environment | | Availability | | Group |
| Ambient (environmental temperature) | 1 | Physical Environment | Accessibility | Availability | | Group |
| Birthplace (inside vs outside US) | 1 | Physical Environment | Accessibility | | | Individual |
| Calendar month | 1 | Physical Environment | Accessibility | | | Individual |
| Patterns of food source destinations | 1 | Physical Environment | | Availability | | Group |
| Home damage | 1 | Physical Environment | Accessibility | Availability | Utilization | Both |
| Relocation | 1 | Physical Environment | Accessibility | Availability | Utilization | Both |
| Disaster assistance | 1 | Physical Environment | Accessibility | Availability | Utilization | Both |
| Spatial access | 1 | Physical Environment | Accessibility | Availability | | Both |
| Transportation mode | 1 | Physical Environment | Accessibility | | | Individual |
| Shopping distance | 1 | Physical Environment | Accessibility | Availability | Utilization | Both |
| Member of social or civic organization | 1 | Social Environment | Accessibility | | | Individual |
| Personal disparity | 1 | Social Environment | Accessibility | | | Individual |
| Number of people in social network | 1 | Social Environment | Accessibility | | Utilization | Individual |
| Church (community characteristic) | 1 | Social Environment | Accessibility | Availability | | Both |
| Neighborhood participation index | 1 | Social Environment | Accessibility | | Utilization | Group |
| Neighborhood social cohesion | 1 | Social Environment | Accessibility | | Utilization | Group |
| Neighborhood problems index | 1 | Social Environment | Accessibility | Availability | | Group |
| Lifetime racial discrimination | 1 | Social Environment | Accessibility | | | Individual |
| Neighborhood congruence | 1 | Social Environment | Accessibility | | | Group |
| Neighborhood SES | 1 | Social Environment | Accessibility | Availability | | Group |
| Neighborhood race/ethnic statuses | 1 | Social Environment | Accessibility | Availability | | Group |
| Sense of community | 1 | Social Environment | Accessibility | | Utilization | Group |
| SNAP policy change | 1 | Social Environment | Accessibility | | | Group |

similar themes could lead to distinct differences between metrics of food security. The authors of this scoping review encourage researchers to utilize standardized metrics, in addition to any questionnaire modification they desire, so that the body of work has a standard for comparison. Efforts such as the Core Outcome Measures in Effectiveness Trials Initiative (COMET) have been working towards standardizing outcomes as a means of reducing research wastage [43]. The rationale for using standard outcomes is that this approach facilitates comparison between studies. Inclusion of a standard outcome, like the USDA 18-item questionnaire, is not a barrier to adding additional outcomes that researchers are interested in investigating.

**Table 5. Health-related risk factors mapped to the dimensions of food security.**

| Term | Citation Hits | Sub Category | Accessibility | Availability | Utilization | Level |
|---|---|---|---|---|---|---|
| Health insurance status | 4 | Physical Health | Accessibility | | Utilization | Both |
| Impairment that limits/prevents use of public transportation | 2 | Physical Health | Accessibility | | Utilization | Individual |
| Alcoholism | 2 | Mental Health | Accessibility | | Utilization | Individual |
| Cancer type | 1 | Physical Health | Accessibility | | Utilization | Individual |
| Time since cancer diagnosis | 1 | Physical Health | Accessibility | | Utilization | Individual |
| Difficulty walking | 1 | Physical Health | Accessibility | | | Individual |
| Difficulty sitting | 1 | Physical Health | Accessibility | | | Individual |
| Difficulty standing | 1 | Physical Health | Accessibility | | Utilization | Individual |
| Difficulty lifting/carrying (10lbs) | 1 | Physical Health | Accessibility | | Utilization | Individual |
| Length of time on dialysis | 1 | Physical Health | Accessibility | | Utilization | Individual |
| HIV status | 1 | Physical Health | Accessibility | | Utilization | Individual |
| Arthritis | 1 | Physical Health | Accessibility | | Utilization | Individual |
| Joint pain | 1 | Physical Health | Accessibility | | Utilization | Individual |
| Liver fibrosis | 1 | Physical Health | Accessibility | | Utilization | Individual |
| Mastery score | 1 | Mental Health | Accessibility | | Utilization | Individual |

Results from this scoping review also suggest that the three unique dimensions of food security (food accessibility, availability, and utilization) are represented by distinct risk factor categories across the peer-reviewed literature and are not equally evaluated by authors. It is critical for researchers to acknowledge that risk factors linked to food accessibility have received more risk factor representation across the published literature and that other dimensions of food security, food availability and food utilization, must be explored to better serve African American adults who experience barriers linked to food insecurity.

Another gap includes the absence of synthesized results for risk factors that have received the most study representation across the peer-reviewed literature. Multiple demographic risk factors including education, age, race/ethnicity, and gender were assessed for an association with food insecurity among most of the included studies. Currently, there is a potential to conduct systematic reviews on extensively evaluated demographic risk factors (age, gender, and race/ethnicity) and summarize the associations found across populations. A systematic review of these risk factors might expose which demographic factors are associated with the highest risk of food insecurity among African American adults in the United States.

Another characteristic includes the frequent use of cross-sectional study designs (n = 73) compared to cohort or longitudinal study designs (n = 19) and randomized controlled trials (n = 3). As noted by multiple authors of the included studies, the use of the cross-sectional design limits the assertion of a causal relationship between exposure variables and outcomes of interest [24]. However, there is an opportunity to consider the implementation of other designs such as cohort study designs. The value that could be obtained from studying groups that do not experience food insecurity and then become food insecure would eliminate many

**Table 6. COVID-19 related risk factors mapped to the dimensions of food security.**

| Term | Citation Hits | Sub-category | Accessibility | Availability | Utilization | Level |
|---|---|---|---|---|---|---|
| Impact of COVID-19 on Employment | 1 | COVID-19 | Accessibility | | Utilization | Individual |
| State stay-at-home orders | 1 | COVID-19 | Accessibility | | Utilization | Group |
| Decreased income (COVID-19) | 1 | COVID-19 | Accessibility | | Utilization | Both |
| Unemployed (prior to pandemic) | 1 | COVID-19 | Accessibility | | Utilization | Individual |

of the limitations of trying to understand the cause and effect presented across the peer-reviewed literature.

## Limitations

The focus of this scoping review was on peer-reviewed literature, and it is unclear if inclusion of grey literature would have impacted review findings.

## Conclusions

The findings from this scoping review suggest that metrics of food security and risk factors associated with food insecurity among African American adults have received variable levels of representation across the literature. The implementation of standardized metrics of food insecurity across the literature would minimize research wastage and facilitate better comparisons between studies. In addition, it is critical for researchers to consider the wide range of food security metrics that are implemented by authors and how the creation of new metrics or modification of standardized metrics could impact the ability to synthesize findings in this critical area. It is also crucial that researchers consider extensively studied risk factors that are eligible for systematic reviews (education, age, race/ethnicity, and gender) as they consider current data gaps and next steps required to address them. For example, behavioural risk factors and risk factors mapped to the food availability dimension of food security require further investigation to better assess human behaviour and environmental factors linked to food availability, and barriers that impact African American populations in the United States. The evaluation of human behaviour and risk factors linked to food availability, a consistent source of quality food, could minimize existing data gaps and the impact of food insecurity as a negative health outcome. Other underrepresented risk factors to consider for future research include factors linked to health disparities among African American adults: lifetime racial discrimination, neighbourhood grocery store availability, neighbourhood safety from violence, income insecurity, and the impact of COVID-19 on employment. For example, households that experience income insecurity or fall below the federal poverty line have greater odds of experiencing inability to afford food, housing insecurity, and food insecurity during the COVID-19 pandemic [41]. In addition, interventions to increase food access among minority and low-income individuals are crucial to minimize health disparities and the economic stress linked to the COVID-19 pandemic [42]. Overall, it is crucial for researchers and clinicians to consider the impact of these factors and how they relate to forms of systemic racism, food insecurity, and the COVID-19 pandemic in the United States.

## Supporting information

**S1 Appendix. Search strategies.**
(DOCX)

**S2 Appendix. Characteristics of included studies.**
(DOCX)

**S3 Appendix. Screening forms.**
(DOCX)

**S1 Checklist. Preferred Reporting Items for Systematic reviews and Meta-Analyses extension for Scoping Reviews (PRISMA-ScR) checklist.**
(PDF)

**S1 File.**
(DOCX)

## Author Contributions

**Conceptualization:** Elizabeth Dennard, Elizabeth Kristjansson, Donna Winham, Annette O'Connor.

**Data curation:** Elizabeth Dennard, Nedelina Tchangalova, Sarah Totton.

**Investigation:** Elizabeth Dennard, Sarah Totton.

**Supervision:** Annette O'Connor.

**Writing – original draft:** Elizabeth Dennard, Annette O'Connor.

**Writing – review & editing:** Elizabeth Dennard, Elizabeth Kristjansson, Nedelina Tchangalova, Sarah Totton, Donna Winham, Annette O'Connor.

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
