## [Decision Letter · Decision Letter 0]

31 May 2022

PONE-D-21-20858Food Insecurity among African Americans in the United States: An evidence and gap mapPLOS ONE

Dear Dr. Dennard,

Thank you for submitting your manuscript to PLOS ONE. After careful consideration, we feel that it has merit but does not fully meet PLOS ONE’s publication criteria as it currently stands. Therefore, we invite you to submit a revised version of the manuscript that addresses the points raised during the review process.

We look forward to receiving your revised manuscript.

Kind regards,

Volkan Okatan

Academic Editor

PLOS ONE

Journal Requirements:

2. Please include a separate caption for each figure in your manuscript.

3. Please identify your study as a scoping review in your title.

Additional Editor Comments:

Dear Authors, we have 2 major revisions for your paper. Please correct demands of reviewers and upload back. When we get your correct paper, we will send reviewers again.

Reviewers' comments:

Reviewer's Responses to Questions

**Comments to the Author**

1. Is the manuscript technically sound, and do the data support the conclusions?

Reviewer #1: Yes

Reviewer #2: Partly

2. Has the statistical analysis been performed appropriately and rigorously? 

Reviewer #1: N/A

Reviewer #2: N/A

3. Have the authors made all data underlying the findings in their manuscript fully available?

Reviewer #1: Yes

Reviewer #2: Yes

4. Is the manuscript presented in an intelligible fashion and written in standard English?

Reviewer #1: Yes

Reviewer #2: No

5. Review Comments to the Author

Reviewer #1: This is an interesting paper that conducts a review of the literature on the food insecurity of Black non-Hispanic households in order to identify describe the full range of the factors that are affecting food insecurity (and being affected by food insecurity) as well as identifying patterns. I think the paper would benefit from a few changes. I remain on-the-fence about the length of the tables. They are repetitive and at the same time informative. I encourage the authors to continue to think about how to best present this information.

A common issue throughout the manuscript is a lack of definition and explanation of key terms. For example, unless I missed it “food insecurity” is not defined. The same holds for the various dimensions of food insecurity, like food availability & accessibility. What are the “proxy variables” for food insecurity (p.6) and how were they determined? Also, in Table 7 how did the authors identify how the “group” was Black? Many datasets only include information on the respondent, how then was a group identified as Black from that information?

In Table 4, I’m wondering about the helpfulness of directly quoting the “authors’ definition of the food security metric.” Would it not be better for the authors of this paper to distill this information and describe it? For example, the very first entry includes phrases like “kappa coefficient,” validity, sensitivity, and birth certificate. These are not definitions of food security. How is breastfeeding initiation a measure of food insecurity? There needs to be some justification for this extra column and why readers need more information than is supplied in the 2nd column.

Table 5 seems a bit careless to me. Some of the articles use NHANES data, which is nationally representative. So, there won’t be “state” or “region,” so “not reported” is not helpful. Why would it not be “All 50 US states + DC” like with Chakrabarti et al. 2021? There’s simply no state or region to be reported. For something like “concept mapping,” it is unclear how/what/why that is a study design.

The authors need to justify the inclusion of Table 6. So much “not reported” undermines any helpfulness of this table.

Perhaps I missed it, but do the authors explain the “sub-category” labels they use in Table 7? Also, columns 4-6 are repetitive. Why not use a “dimension” heading and then use accessibility, availability, utilization. It seems most rows have just “accessibility.” Abbreviations could be used to fit them all into one column. How was individual versus group determined?

Several conclusions in the Discussion section need support. The authors state that because demographic & environmental categories (where these ever defined?) represent the greatest number of risk factors, they have “adequate representation.” How is that? Another one is “inference obtained from a single estimate is limited.” What does that mean? How are accessibility, availability, utilization “hierarchical dimensions?” This section needs a careful revision.

Other issues:

- The protocol (p.4) is listed using a long title. But what is it and how does it work? The authors needs to carefully describe their methods.

- Why were these 6 databases chosen (p.6)? Why not Google Scholar as well?

- I don’t understand some of the query terms in Table 1. What is “#3 not #4”?

- The first two subsections on p.12 need clarifying. The direct quotation of Munn and colleagues is awkward.

Reviewer #2: The idea and objectives are interesting. Howeever, the process and indication of findings are poor. Actually, I did not fully understand the relationship proposed for COVID-19 process relying on the previous literature. But more impotantly, detailed layout of previous literature which is hard top follow up and inefficient comparison relying on the citation number or number of participants in relevant research outputs disables the readerr to understand the correleation between objectives and findings. Besides, findings and discussion is rather poor with reference to the layout. I suggest serios overview of the layout and findigns & discussion. Besides, the paper should be shortened via objective-relevant referencing to the previous literature. With its apparent content and form, the paper is not eligible for publication.

6. PLOS authors have the option to publish the peer review history of their article (what does this mean?). If published, this will include your full peer review and any attached files.

Reviewer #1: No

Reviewer #2: No

---

## [Author Response · Author response to Decision Letter 0]

6 Jul 2022

Dear Editors, 

Thank you so much for giving us the opportunity to submit a revised draft of our manuscript, Food insecurity among African Americans in the United States: A scoping review, for publication at PLOS ONE. We appreciate the time and effort that reviewers dedicated to this rigorous review process. All co-authors have carefully considered each insightful comment and incorporated most reviewer suggestions to strengthen the latest version of our manuscript. 

All responses to reviewer comments are provided below, while manuscript modifications are highlighted in red text via tracked changes. 

We look forward to hearing from you all regarding our submission and next steps required for publication. 

Please let me know if you need more information.

Thank you so much for your time and patience, 

Elizabeth Dennard, MPH

Elizabeth.dennard@fda.hhs.gov

Response to Reviewer #1:

Reviewer Comment: This is an interesting paper that conducts a review of the literature on the food insecurity of Black non-Hispanic households in order to identify describe the full range of the factors that are affecting food insecurity (and being affected by food insecurity) as well as identifying patterns. I think the paper would benefit from a few changes. I remain on-the-fence about the length of the tables. They are repetitive and at the same time informative. I encourage the authors to continue to think about how to best present this information.

Author Response: Thank you so much for your detailed comments and constructive feedback. Absolutely, we agree that multiple tables would benefit from a few changes and should be moved to the Supplementary Materials. We moved Table 1 – 3 (Revised Titles: Table S1 – S3) to Appendix A and Table 4 – 6 (Revised Titles: Table S4 – S6) to Appendix B so these tables do not crowd the manuscript and readers can readily reference these items under the Supplementary Materials. 

Reviewer Comment: A common issue throughout the manuscript is a lack of definition and explanation of key terms. For example, unless I missed it “food insecurity” is not defined. The same holds for the various dimensions of food insecurity, like food availability & accessibility. What are the “proxy variables” for food insecurity (p.6) and how were they determined?

Author Response: Absolutely, this is an excellent point. We created a glossary of food security terms and proxy variables (Table 1), so definitions are readily available to readers (pg. 4). Also, food accessibility, food availability, and food utilization serve as proxy variables and unique dimensions of food insecurity. 

Reviewer Comment: Also, in Table 7 how did the authors identify how the “group” was Black? Many datasets only include information on the respondent, how then was a group identified as Black from that information?

Author Response: Thank you so much for your constructive feedback. Table 7 includes demographic risk factor mapping details while Table S6 includes the spread of participant ages and participant-household counts. If Reviewer #1 is referring to Table S6 in the comment above, these participant and household counts were included or explicitly described in each reference. If an author did not include an age limit or number of participants-households, reviewers entered “not reported” into the table. Please let us know if there is another comment linked to Table 7 and we will address this concern as soon as possible. 

Reviewer Comment: In Table 4, I’m wondering about the helpfulness of directly quoting the “authors’ definition of the food security metric.” Would it not be better for the authors of this paper to distill this information and describe it? For example, the very first entry includes phrases like “kappa coefficient,” validity, sensitivity, and birth certificate. These are not definitions of food security. How is breastfeeding initiation a measure of food insecurity? There needs to be some justification for this extra column and why readers need more information than is supplied in the 2nd column.

Author Response: Absolutely, this is an excellent point. Table S4 illustrates the number of definitions or metrics (n = 37) used to assess food insecurity across the 98 references included in our review. Reviewers included the definition of food insecurity described by each reference to demonstrate how the metric was identified in the text. We incorporated your suggestion by moving Table S4 to the Supplementary Materials under Appendix B. 

Reviewer Comment: Table 5 seems a bit careless to me. Some of the articles use NHANES data, which is nationally representative. So, there won’t be “state” or “region,” so “not reported” is not helpful. Why would it not be “All 50 US states + DC” like with Chakrabarti et al. 2021? There’s simply no state or region to be reported. 

Author Response: This is an excellent point. During the review process, reviewers captured the state and region of each reference if it was described by authors in the text. After reviewing references with a “not-reported” submission under the “state(s)” column in Table S5, we can confirm that these citations did not reference NHANES data. Also, Morales et al., 2020 explicitly describes the study location as “All 50 US states + DC” while other citations do not include this information. 

Reviewer Comment: For something like “concept mapping,” it is unclear how/what/why that is a study design.

Author Response: Absolutely, this is an excellent point. A concept map serves to visually display ideas and relationships between described concepts. We confirmed that the term “concept mapping” was reported by Barnidge et al., 2017 as a study design in the text. 

Reviewer Comment: The authors need to justify the inclusion of Table 6. So much “not reported” undermines any helpfulness of this table.

Author Response: Thank you for your constructive suggestion. We believe that the study characteristics presented in Table S6 (age, African American study participant counts, and household counts) are relevant findings that should be accessible to readers and other reviewers. However, we agree that Table S6 includes many “not-reported” entries. We moved Table S6 to Appendix B to minimize manuscript crowding. 

Reviewer Comment: Perhaps I missed it, but do the authors explain the “sub-category” labels they use in Table 7? Also, columns 4-6 are repetitive. Why not use a “dimension” heading and then use accessibility, availability, utilization. It seems most rows have just “accessibility.” Abbreviations could be used to fit them all into one column. How was individual versus group determined?

Author Response: Absolutely, these are excellent points. Sub-category or proxy variable definitions are described throughout the manuscript, but revisions include a glossary of terms to readily inform readers (pg. 4). We believe that including three separate columns for food access, availability, and utilization illustrates the number of gaps presented across the literature. Though we appreciate this suggestion, we contest this revision because it could minimize the visual differences between food security dimensions presented in Table 7 – 11 (Revised titles: Table 2 – 6).

Reviewer Comment: Several conclusions in the Discussion section need support. 

Author Response: This is an excellent point. We revised the discussion section of our manuscript, so our findings are clearer, and our conclusion is properly supported by our claims and included references.

Reviewer Comment: The authors state that because demographic & environmental categories (where these ever defined?) represent the greatest number of risk factors, they have “adequate representation.” How is that? 

Author Response: Absolutely, this is an excellent point. We adjusted the wording from “adequate representation” to “more representation” to illustrate the lack of representation for the remaining categories. This section highlights that some categories have received more attention or representation across the literature while others have been neglected or received less attention by authors.

Reviewer Comment: Another one is “inference obtained from a single estimate is limited.” What does that mean? 

Author Response: Thank you so much for your inquiry. The inference of a single estimate section is in reference to the lack of behavioral risk factors assessed by authors. We adjusted the sentence structure above to make our assertion clearer. “The inference obtained from a single publication is limited; therefore, authors of future studies should consider findings from multiple studies to refine metrics and improve study design for stronger inference about described associations (line 371 - 375).”

Reviewer Comment: How are accessibility, availability, utilization “hierarchical dimensions?” This section needs a careful revision.

Author Response: This is an excellent point. The term “hierarchical” may cause confusion for readers by implying that food access, availability, and utilization are arranged in order of rank but for the purpose of our manuscript, they are categories that interact with one another and impact individuals in unique ways. We adjusted the wording from “hierarchical” to “unique” to illustrate the term differences. 

Other issues:

Reviewer Comment: The protocol (p.4) is listed using a long title. But what is it and how does it work? The authors need to carefully describe their methods. 

Author Response: Absolutely, this is an excellent point. We provided more information linked to the purpose of the PRISMA-ScR checklist under Materials and Methods (Protocol and Registration). 

Reviewer Comment: Why were these 6 databases chosen (p.6)? Why not Google Scholar as well?

Author Response: Thank you so much for pointing this out. In addition to database searches, Google Scholar was searched to find any additional studies that may have been missed through the database searches. We included this clarification in the revised manuscript under Search sources. 

Reviewer Comment: I don’t understand some of the query terms in Table 1. What is “#3 not #4”?

Author Response: This is an excellent point. We reformatted Table 1 (Revised Title: Table S1) and added a column explaining the actions for each step of the database searches. Also, we moved Tables 1 – 3 (Revised Titles: Table S1 – S3) to Appendix A under Supplementary Materials.

Reviewer Comment: The first two subsections on p.12 need clarifying. The direct quotation of Munn and colleagues is awkward.

Author Response: Thank you so much for your constructive feedback. We did not assess risk of bias or study quality of the included studies, as risk-of-bias assessment is not required for scoping reviews. We adjusted our justification and the direct quotation from Munn and colleagues to make our justification for excluding risk of bias and study quality clearer (line 189 - 194).

Response to Reviewer #2: 

Reviewer Comment: The idea and objectives are interesting. However, the process and indication of findings are poor. I did not fully understand the relationship proposed for COVID-19 process relying on the previous literature. 

Author Response: Thank you so much for your constructive feedback. In 2021, PLOS ONE editors encouraged us to update our search findings. We decided to update our search (1st search: 11/19/19; 2nd search: 5/20/21) and include relevant findings in the latest version of our manuscript (submitted July 2021). Also, COVID-19 findings or the COVID-19 risk factors identified (n = 4) do not rely on previous search findings (1995 to 2019) because COVID-19 related references were published between 2020 and 2021.

Reviewer Comment: But more importantly, detailed layout of previous literature which is hard to follow up and inefficient comparison relying on the citation number or number of participants in relevant research outputs disables the reader to understand the correlation between objectives and findings. 

Author Response: Thank you for your constructive feedback. We believe that the characteristics presented in Table S5 (study characteristics) and Table S6 (population characteristics) are relevant findings that should be accessible to readers and other reviewers. We moved Table 4 - 6 (Revised Titles: Table S4 – S6) to Appendix B so these items do not crowd the manuscript and readers can readily access them under the Supplementary Materials. 

Reviewer Comment: Besides, findings and discussion are rather poor with reference to the layout. I suggest serios overview of the layout and findings & discussion. Besides, the paper should be shortened via objective-relevant referencing to the previous literature. With its apparent content and form, the paper is not eligible for publication.

Author Response: This is an excellent point. We revised the discussion and conclusion section of our manuscript, so our findings are clearer and supported properly.

Reviewer Comment: While revising your submission, please upload your figure files to the Preflight Analysis and Conversion Engine (PACE) digital diagnostic tool, https://pacev2.apexcovantage.com/. PACE helps ensure that figures meet PLOS requirements. To use PACE, you must first register as a user. Registration is free. Then, login and navigate to the UPLOAD tab, where you will find detailed instructions on how to use the tool. If you encounter any issues or have any questions when using PACE, please email PLOS at figures@plos.org. Please note that Supporting Information files do not need this step.

Author Response: Absolutely, we uploaded our figures to PACE, which converted our PDFs to tif files, via the PLOS ONE editorial manager site in 2021. 

Other Issues: 

Reviewer Comment: Covid – racial discrimination? (line 61)

Author Response: Thank you so much for your inquiry linked to the relationship between racial discrimination and the impact of COVID-19. Since the start of the pandemic, African American adults have experienced more negative health outcomes linked to COVID-19 than other populations due to racial discrimination, income disparities, and inconsistent access to food [2]. This reference highlights the need to address COVID-19 related risk factors linked to food insecurity by exposing health disparities based on factors linked to one’s race and ethnicity. 

Reviewer Comment: 1995 – 2019 literature and potential covid linkage is weak (line 418)

Author Response: We are not claiming that literature published between 1995 and 2019 is linked to the impact of COVID-19 in 2020 or 2021. We updated our search last year to include studies published between 1995 and May 20, 2021. This update captured citations that addressed COVID-19 related risk factors (impact of COVID-19 on employment, state stay-at-home orders, decreased income due to COVID-19, and unemployment prior to the pandemic) and how they are linked to food insecurity as an outcome.

---

## [Decision Letter · Decision Letter 1]

30 Aug 2022

Food Insecurity among African Americans in the United States: A scoping review

PONE-D-21-20858R1

Dear Dr. Dennard,

We’re pleased to inform you that your manuscript has been judged scientifically suitable for publication and will be formally accepted for publication once it meets all outstanding technical requirements.

Kind regards,

Volkan Okatan

Academic Editor

PLOS ONE

Additional Editor Comments (optional):

Reviewers' comments:

Reviewer's Responses to Questions

**Comments to the Author**

1. If the authors have adequately addressed your comments raised in a previous round of review and you feel that this manuscript is now acceptable for publication, you may indicate that here to bypass the “Comments to the Author” section, enter your conflict of interest statement in the “Confidential to Editor” section, and submit your "Accept" recommendation.

Reviewer #1: All comments have been addressed

2. Is the manuscript technically sound, and do the data support the conclusions?

Reviewer #1: Yes

3. Has the statistical analysis been performed appropriately and rigorously? 

Reviewer #1: N/A

4. Have the authors made all data underlying the findings in their manuscript fully available?

Reviewer #1: Yes

5. Is the manuscript presented in an intelligible fashion and written in standard English?

Reviewer #1: Yes

6. Review Comments to the Author

Reviewer #1: The authors addressed my comments and I think the manuscript is improved.

The minimum character count has now been met.

7. PLOS authors have the option to publish the peer review history of their article (what does this mean?). If published, this will include your full peer review and any attached files.

Reviewer #1: No

---

## [Editor Report · Acceptance letter]

2 Sep 2022

PONE-D-21-20858R1 

Food insecurity among African Americans in the United States: A scoping review 

Dear Dr. Dennard:

I'm pleased to inform you that your manuscript has been deemed suitable for publication in PLOS ONE. Congratulations! Your manuscript is now with our production department. 

Kind regards, 

on behalf of

Dr. Volkan Okatan 

Academic Editor

PLOS ONE